

# Technical note: A global database of the stable isotopic ratios of meteoric and terrestrial waters

Annie L. Putman[1,2] and Gabriel J. Bowen[1,2]

[1]University of Utah Geology & Geophysics, University of Utah, 383 F.A. Sutton Bldg. 115 S. 1460 E., Salt Lake City, UT 84112-0102, US
[2]Global Change & Sustainability Center, University of Utah, 234 F.A. Sutton Bldg. 115 S. 1460 E., Salt Lake City, UT 84112-0102, US

**Correspondence:** Annie Putman (putmanannie@gmail.com)

**Abstract.** The hydrogen and oxygen stable isotope ratios of water have been used to identify sources, transport pathways, and phase-change processes within the water cycle, supporting hydrologic, forensic, ecologic, and hydroclimatic investigations. Here, we introduce an unique, open-access, global database of stable water isotope ratios ($\delta^{18}O$, $\delta^{17}O$ and $\delta^{2}H$) from various waters types. This database facilitates data preservation, supports standardized metadata collection, and decreases the time
investment for metanalytic research and reference dataset discovery. As of April 2019, the database includes 227,699 samples from 51,321 sites, associated with 207 projects, spanning 1949 through 2019. Key information stored includes the hydrogen and oxygen isotope ratios, water type, collection date and time, site location, and project information. To promote rapid data discovery and collaboration, the database exposes metadata and data owner contact information embargoed data, but only permits downloads of public data. The database is supported by two companion apps, one for processing and upload of analytical
data from laboratories and the other an iOS application that supports digital collection of sample metadata.

## 1 Introduction

Stable isotopes of hydrogen and oxygen in terrestrial and plant waters have proven useful for addressing questions of ecohydrologic connectivity, residence time changes, and fluxes between the atmospheric and continental branches of the hydrologic
cycle (e.g., Jasechko et al., 2013; Brooks et al., 2010; Ala-aho et al., 2018; Bowen et al., 2018). For example, terrestrial waters, like lakes, rivers, soil waters, and groundwaters, exhibit complex connectivity to precipitation and among terrestrial water pools (Good et al., 2015), and natural variation in water isotopes, if well documented through reference data, can be used to link sampled water to its sources. In atmospheric branch of the hydrologic cycle, water isotopes have been used to identify continental and oceanic sources of vapor and the lateral and vertical transport of water by circulation systems (e.g., Aemisegger
et al., 2015; Fiorella et al., 2015; Cai and Tian, 2016). Data documenting both the continental and atmospheric domains inform



each other and research into forensic questions like food or water provenance (Oerter et al., 2017; Jameel et al., 2018) and ecohydrologic questions like plant water use (Oerter et al., 2019).

Oxygen has three major isotopes, $^{16}O$ (99.757%), $^{17}O$ (0.038%) and $^{18}O$ (0.205%), and hydrogen has two stable isotopes $^{1}H$ (99.9885%) and $^{2}H$ (0.0115%). The isotopologues most often measured in water are $^{1}H_2^{16}O$, $^{1}H^2H^{16}O$, and $^{1}H_2^{18}O$, with fewer $^{1}H_2^{17}O$ measurements. Isotopologue abundance is reported as the Vienna Standard Mean Ocean Water (VSMOW)-normalized heavy-to-light isotope ratio ($\delta = \frac{R_{SA}-R_{VSMOW}}{R_{VSMOW}}$, where $R = \frac{^{18}O}{^{16}O}$, $\frac{^{17}O}{^{16}O}$ or $\frac{^{2}H}{^{1}H}$), in ‰ (Coplen et al., 1996).

Variability in $\delta^{18}O$, $\delta^2H$, and $\delta^{17}O$ arise from isotope fractionation during water cycle processes. The term fractionation refers to the sorting of heavy atom ($^{18}O$, $^{17}O$ or $^{2}H$)-carrying water molecules from the water molecules comprised of only light atoms ($^{16}O$ and $^{1}H$), and occurs during phase changes (e.g., evaporation, condensation, and deposition) and across humidity gradients (e.g., vapor diffusion from a saturated water surface into dry air above (Craig and Gordon, 1965)). Fractionation factors, which quantify the strength of sorting, are controlled by both temperature and humidity, where the isotopic sorting effect increases at cooler temperatures and for strong vapor pressure gradients.

The majority of the meridional and altitudinal variation in $\delta$ values of observed meteoric waters (precipitation) arises from variation in the extent of Rayeligh distillation: the progressive rainout of heavy isotopologues during the evolution of a precipitating airmass (Gat, 1996). The spatial and temporal variability in precipitation $\delta$ values arising from water cycle processes imprints on terrestrial and ecological water pools. Most hydrologic processes (e.g, infiltration, evapotranspiration) reflect mixing of different source waters as opposed to fractionation. Thus, precipitation isotope ratios provide a framework for interpreting variability observed in terrestrial and ecologic water pools. For example, precipitation and terrestrial waters can be used together in mixing models to estimate groundwater provenance (Jasechko et al., 2014) and lake water recharge season (Bowen et al., 2018). Likewise, water isotopes can trace the seasonality of precipitation utilized by trees and plants (Brooks et al., 2010) and the origin of food or drinking water (Oerter et al., 2017; Jameel et al., 2018). Large regional collections of precipitation data have been interpreted in terms of hydroclimatic variability (Liu et al., 2010).

Pioneering research in the areas of climate, hydrology, ecology and forensics that use water isotopes have relied upon large continental or global datasets (Dansgaard, 1964; Rozanski et al., 1992; Bowen and Wilkinson, 2002; Liu et al., 2010), or have assimilated multiple datasets (Masson-Delmotte et al., 2008; Jasechko et al., 2013; Li and Garzione, 2017). These projects highlight the types of research that might be supported by an organized, publicly available archive of oceanic, meteoric, and terrestrial water isotope datasets. Furthermore, such an archive would allow researchers to identify data availability and the extent of previous work in their research area and provide a centralized location for data archiving to satisfy the requirements of funding agencies. A publicly available database might also be used as an educational tool for instruction or class projects (Oerter et al., 2017).

Currently, there are four publicly available water isotope databases. The Stable Water Vapour Isotope Database (SWVID) (Wei et al., 2019), which stores timeseries data for water vapor sampled at several dozen sites worldwide, is hosted by Yale University and supported by the U. S. National Science Foundation. The Global Seawater Database (Schmidt et al., 1999), is a periodically-updated collection of ocean water isotope data. The International Atomic Energy Agency manages two databases





that largely serve to distribute data from their long-standing isotope monitoring programs, the Global Network of Isotopes in Precipitation (GNIP) and the Global Network of Isotopes in Rivers (GNIR) archives (IAEA/WMO, 2019).

In this paper we describe the Waterisotopes Database, the 'wiDB', a relational database that archives stable isotope data for a wide range of environmental waters sampled at sub-daily to multiyear temporal scales. The database was originally developed as a private resource supporting the development of gridded precipitation isotope data products (Bowen and Wilkinson, 2002; Bowen et al., 2005), and primarily contained measurements from the PI's lab and precipitation isotope data from GNIP. Over time, it has grown in scope through contributions by the community (e.g., Mayer, 2016; Csank, 2017; Nelson, 2018; Thomas, 2018), and assimilation of literature data (e.g., Xie et al., 2011; Yao et al., 2013; Oshun et al., 2016). About two years ago we began developing open access protocols for the wiDB and publicizing it as a community archive. The current goal for the project is to develop a community repository providing a comprehensive compilation of stable isotope water data via a plat-form supporting and promoting Findable, Accessible, Interoperable, Reusable (FAIR) data management practices (Wilkinson et al., 2016). Our hope is that the wiDB will both facilitate and improve data preservation within the water isotope research community, in part through the structured archive process, and in part through standardizing the metadata recorded alongside samples in future studies. Web and programmatic exposure of metadata for all wiDB data will support data discovery and sharing within the community, hopefully expanding the accessibility and scope of water isotope-enabled science.

## 2  Methods

The wiDB is a relational MySQL database hosted by the University of Utah Center for High Performance Computing, and available without authentication via a web search interface and custom APIs. A public API is in development. The wiDB hosts a wide variety of environmental water types, including *Precipitation*, *Rime*, *Lake*, *River_or_stream*, *Ocean*, *Ground*, *Soil*, *Stem*, *Cave_drip*, *Mine*, *Spring*, *Tap*, *Bottled*, *Sprinkler*, *Canal*, *Snow_pit*, *Firn_core*, *Ice_core*, *Cloud_or_fog* and *Vapor*. The wiDB has five tables, *Water_Isotope_Data*, *Samples*, *Sites*, *Projects*, and *Climate_Data* (Figure 1). The *Samples* table contains the unique *Sample_ID*, the sample start and collection dates, and the water type. The *Samples* table is linked to the *Sites* table, which stores site metadata like latitude, longitude, and country, The *Water_Isotope_Data* table, which contains analytical results and metadata, the *Climate_Data* table, which stores basic climate information like temperature and precipitation amount, and the *Projects* table, which records the dataset contributor's contact and data citation information as well as a public/private flag that is used to restrict access to data not (currently) intended for distribution.

We introduce new data to the database in three ways. First, data may be introduced via the calibration and storage procedures associated with in-house water isotope analyses. Data from all water isotope analysis performed at the University of Utah Stable Isotopes Facility for Environmental Research (SIRFER) lab are automatically stored in the wiDB *Water_Isotope_Data* table as a component of the lab's processing and calibration routines. Researchers analyzing their samples in the SIRFER facility are encouraged to provide metadata to accompany the submission of their samples, which are then uploaded along with the analytical data, making those results discoverable, and (if redistributable) usable by others. Although this protocol is currently used only by SIRFIR, we are eager to work with other labs to test and more widely implement similar measurement-to-





archive data protocols. Second, datasets may be contributed by scientists wishing to archive their data to satisfy grant funding requirements or because they support open data initiatives. These datasets are formatted for upload (the template excel file provided to data contributors, 'WI_Template.xlsx' is provided in the supplementary information), checked, and pushed to the database using an R script. Third, a large number of datasets are available as part of peer-reviewed papers or technical

reports. Various such published datasets have been formatted for upload by authors or by other members of the community and uploaded to the database. In some cases, this requires digitizing data (e.g., Jacob and Sonntag, 1991; Scholl et al., 2014), and reconstructing metadata from figures (e.g., Yi et al., 2008).

In an effort to promote standardization of metadata, and to streamline its collection, we have developed and released the wiSamples iOS app. Field metadata can be collected using the app and exported as *.csv files that use the wiDB structure.

Thus, sample metadata captured with the app can be directly imported into the database. The app leverages capabilities like GPS, clock and time zone, and reverse gocoding to autopopulate metadata fields using standardized formats. It also queries the wiDB via an API to display the distribution of existing sampling sites and allows association of new samples with these sites, if appropriate.

## 3  Database Records

As of April 2019, the *Water_Isotope_Data* table contains 248,857 water isotope analyses, 32,027 of which are in-house analyses from the SIRFIR lab. All analyses are from laser instruments (e.g., Picarro or LGR) or isotope ratio mass spectrometers. The water isotope analyses correspond to 227,699 entries in the *Samples* table, distributed among samples types, as shown in Table 1. There are more analyses than samples because some samples are analyzed multiple times, or because some water isotope analyses do not have a matching sample entry in the case that water was analyzed but no metadata were provided. The

samples come from 51,321 sites. Among the 207 projects currently in the database, there are 185 publicly available datasets, 13 proprietary datasets, and 10 datasets that are publicly available elsewhere but for which redistribution is not permitted (e.g., IAEA/WMO, 2015). All projects associated with the database are described individually in the supplemental information.

The countries with the most samples in the database include the United States, China, Canada, and Germany (Figure 2). In general, these countries have both long-term monitoring sites and one or more large-scale spatially distributed sampling

programs, like national studies characterizing the spatial distribution of lake, river, or tap waters.

Aside from *Ice_core* and *Firn_core* samples, the earliest sampling date recorded in the Samples table is in June of 1949, and the most recent is January 2019. In general, most of the samples collected prior to 2000 are *Ocean*, *Precipitation*, *Ground* or *Ice_core* type samples (Figure 3). In part, the temporal bias may reflect the evolution of the science, as a wider range of water types were measured during the expansion of stable isotope applications to ecohydrologic and forensic studies. However, a

30 large amount of data, particularly for groundwater, was collected during earlier timeperiods that hasn't been published, publicly released, or pulled into the wiDB yet. This demonstrates the opportunity for continued work and community involvement in improving this resource.



## 4 Technical Validation

The technical validation methods applied vary for data assimilated in different ways. Samples analyzed at the SIRFIR lab are subject to a set of standard laboratory quality control checks using automated scripts and manual screening before being imported to the database. Samples imported from peer-reviewed publications and other databases are assumed to be quality
controlled as part of the analysis and publication effort. However, these datasets are checked during organization and after upload for reasonable isotope analysis values, dates and times, and geographic locations. Periodically, we manually check the whole database to ensure latitudes between -90° and 90° and longitudes between -180°and 180°. We check for reasonable water isotope analysis values by reference to the Global Meteoric Water Line, which describes the expected linear correlation between $\delta^{18}O$ and $\delta^2 H$, combined with the expected ranges of each isotope in natural samples of various types. If a value is
updated from a prior version, the update is noted in the comments column of the table. Nonetheless, users are advised to refer to original references to cross-check any data that may appear inconsistent with expected values. To support tracing of potential errors, we attempt to record as much site, sample, and analytical metadata as is available and make an effort to provide raw (non-averaged) water isotope analysis values when available.

Among datasets published with manuscript tables or supplements, we have encountered a wide range of metadata complete-
15 ness. Issues include, but are not limited to: missing latitudes and longitudes, geographic data reported in difficult to universalize units (e.g., Township and range system (e.g., Williams and Rodoni, 1997) or localized coordinate systems without necessary reference points), missing sampling start or end data, and/or only including processed data (e.g., precipitation weighted monthly or annual averages) as opposed to raw data. In some cases this may reflect changes in technology(e.g., use of maps vs. handheld GPS units) or lack of community guidelines for metadata completeness. However, in other cases this may reflect an author's
desire to keep certain aspects of a dataset proprietary. In either case, metadata incompleteness reduces the utility of datasets for both metanalyses and as contextual information for related studies. For this reason, we suggest that the water isotope community adopt a systematic standard for completeness in reporting of metadata associated with datasets. This practice will ensure the utility and longevity of our datasets.

## 5 Usage notes and further comments

The database is accessible using a website interface that includes a zoomable-scrollable map showing site locations. The sites are clickable, displaying the sample type(s), number of $\delta^{18}O$ or $\delta^2 H$ analyses ($\delta^{17}O$ are not included yet on the web interface as there are so few data in the wiDB), and range of sample collection dates associated with the site. As well, a link to the data provenance information, including contact names and citations, is provided. A html-based form can be used to search the database using spatial, temporal, sample type, analyte, and project fields. In the future, wiDB access will be provided via a
documented, public API supporting programmatic search and download of data.

In the browser portal interface, all proprietary datasets (either restricted due to the data policy of the contributor (e.g., IAEA/WMO, 2015) or because the authors require a data embargo prior to publication) are present on the map, and all information besides the water isotope analyses can be downloaded. This allows users to 1) discover when and where data have




been collected, even they are not available for direct download and 2) obtain the data owner's contact information, so a user may contact the owner to request access to the dataset. This solution represents a potential incentive for early data archival. It allows data producers to deposit data early in its lifecycle (e.g., immediately after analysis) without compromising their priority access, while also providing exposure for their work that might lead to new collaborations with the potential data-

users and advance the timeline for data discovery and reuse by those users. While allowing and supporting proprietary data is not ideal from the standpoint of FAIR data management practices, the goal reflected in the wiDB design is to recognize that multiple perspectives on timescales for data release persist in the community and offer a middle-ground solution that ensures data archival, but respects the desires of providers for priority use during an embargo period. By exposing metadata, including sampling location, time, and water type, coupled with data owner contact information we hope to promote openness in the

community, drive creative research project design, and facilitate collaboration among researchers.

Since the development of the online database, a private API has been used to pull precipitation data from the database, process them to a common time resolution, and use the resulting dataset as the basis for the Isoscapes Modeling, Analysis and Prediction (IsoMAP) (Bowen et al., 2019) tool, a web-based platform for development of derived, gridded, spatiotemporal isotope data productes (isoscapes). This API has access to all of the data within the database, including private data, but does not

expose any of the data directly to user download. This represents yet another way in which this water isotope database supports and furthers the knowledge base of the broader community of ecologists, forensic scientists, hydrologists and atmospheric scientists who use stable water isotopes.

## 6   Conclusions

In this technical note, we present the global Waterisotopes Database (wiDB), which we have designed as a community repos-

itory that supports FAIR data management practice and provides a comprehensive compilation of stable isotope water data. Key information stored includes the hydrogen and oxygen isotope ratios, water type, collection date and time, site location, and project information. As of April 2019, the database holds 207 projects totaling 227,699 samples from 51,321 sites, spanning the years 1949 through 2019. We hope that the wiDB will improve data preservation within the water isotope research community, in part through the structured archive process, and in part through standardizing the metadata recorded alongside

samples in future studies. In support of those goals, we have developed two associated applications, the first for streamlining the analysis to archiving pipeline, and the second for standardizing and digitizing the sample metadata collection processes. The wiDB web interface is designed to expose metadata for all wiDB data, thereby supporting data discovery and sharing within the community. By introducing and documenting this new set of resources, we hope to expand the accessibility and scope of water isotope-enabled science.

*Code and data availability.*   The R scripts used for SIRFIR data processing and various modes of data upload were developed in-house by the authors and are available via GitHub (specifically 'CRDS_liquid_1.R', 'CRDS_liquid_2.R', 'CRDS_liquid_3.R', 'Upload_metadata.r',





and 'Metadata_functions.r', in https://github.com/SPATIAL-Lab/CRDS-processing). All projects associated with the database are described in the supplemental information. Projects in each subsection are organized by Project ID. All datasets in the list are available from the wiDB unless otherwise noted. If projects are not redistributable, we comment on where data may be accessed.

*Author contributions.* ALP co-wrote the paper, contributed and implemented management ideas, and organized data. GJB co-wrote the

5  paper, initiated the database, developed upload applications, and organized data.

*Competing interests.* The authors declare no competing financial interests.

*Acknowledgements.* This research was supported by NSF grant 1241286, 1565128 and 1759730, and the University of Utah Graduate Research Fellowship. The authors thank Chonghuan Xia for her work on the APIs for wiDB web interface, Karan Sequiera for his design of the web portal, and the Univerity of Utah Center for High Performance Computing for their support of the wiDB. The authors thank Rich

10  Fiorella for his contributions to database discussions and efforts in digitizing and uploading datasets. Finally, the authors thank all of the researchers and organizations who have contributed data.





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




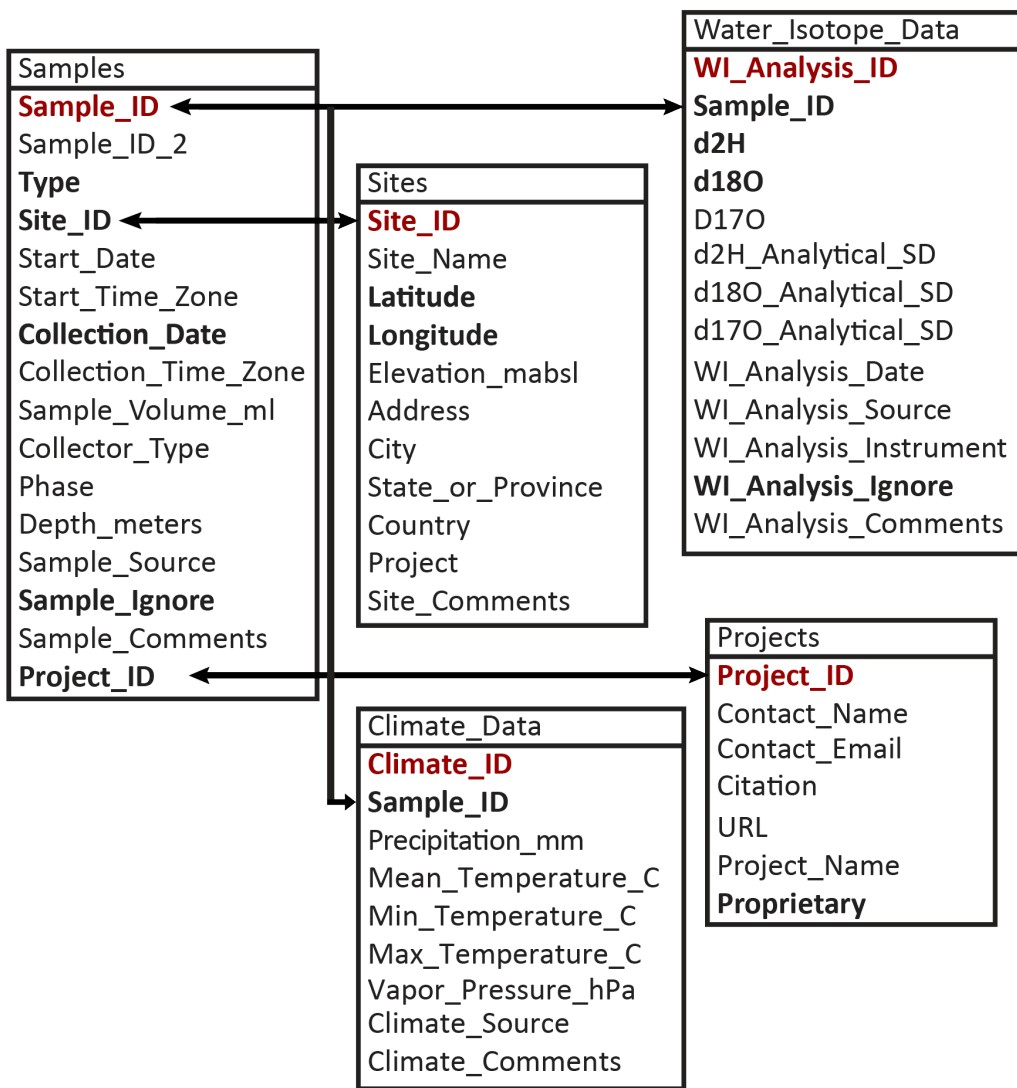

**Figure 1.** The wiDB schema. the 5 data tables are linked to one another using primary keys, highlighted in maroon. Metadata fields required for all entries are bolded.





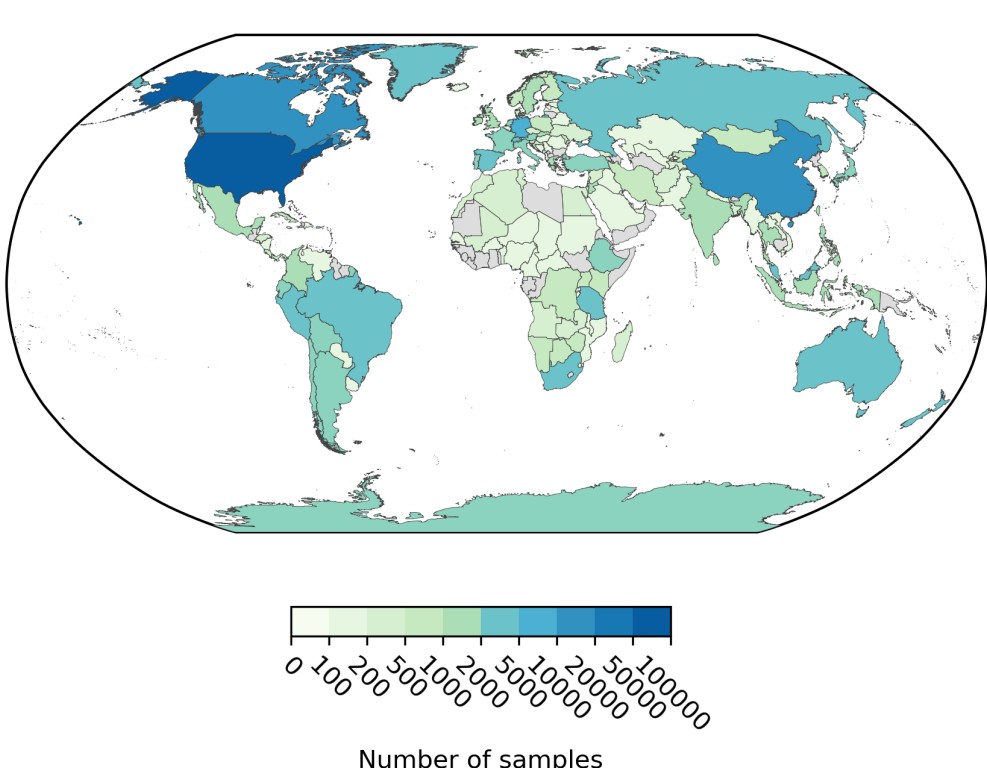

**Figure 2.** Number of samples (of any type) from each country. Countries in gray have no samples currently stored in the database. Note that the segmented color scale is not linear.



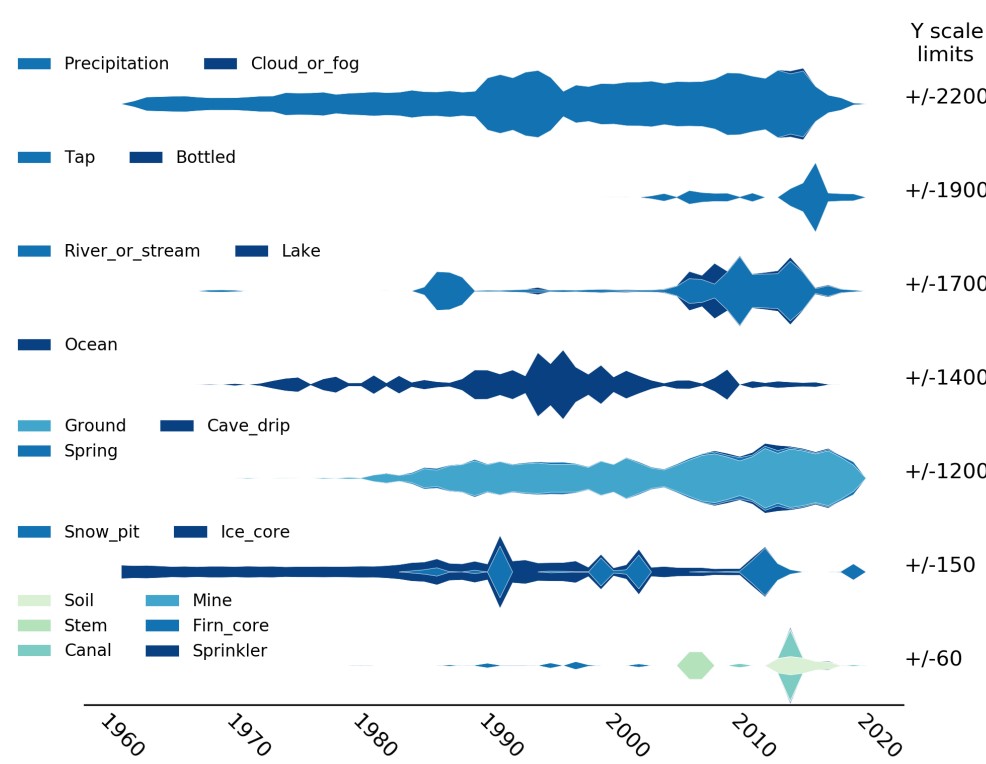

**Figure 3.** Temporal coverage of samples grouped by type and binned by collection year. Plots are ordered by most numerous sample types to most 'niche' sample types, and the y-scale varies by plot. The sample is attributed to the latest year within the collection period, even if it may represent integrated sampling across multiple years (e.g., *Precipitation*). We do not show the entire temporal range of *Ice_core* samples.




**Table 1.** Sample Types and abundance in database.

| Abundance | Type |
|---|---|
| 169 | Bottled |
| 97 | Canal |
| 1257 | Cave_drip |
| 708 | Cloud_or_fog |
| 41 | Firn_core |
| 38637 | Ground |
| 8377 | Ice_core |
| 4730 | Lake |
| 4 | Mine |
| 26693 | Ocean |
| 107894 | Precipitation |
| 47 | Rime |
| 24435 | River_or_stream |
| 1015 | Snow_pit |
| 97 | Soil |
| 3186 | Spring |
| 10 | Sprinkler |
| 84 | Stem |
| 9802 | Tap |
| 374 | Vapor |