# Peer review of "Technical Note: A global database of the stable isotopic ratios of meteoric and terrestrial waters"

_Hydrology and Earth System Sciences, 2019_

## Referee Comment (RC1) · Anonymous Referee #1 · 20 May 2019

The study of sources, transport pathways, and phase-change processes within the water cycle based on hydrogen and oxygen stable isotope of different water types has been carried out in the World. Certainly, sharing and collecting data is a good idea. I really appreciate your works. At the same time, I would like to thank you very much for your efforts in collecting and sharing data on stable isotopes for all water bodies in global. However, there are some points which were not phrased clearly in this technical note, and I have made additional comments below that might be helpful in guiding a revision. Firstly, only the importance of isotope research and the data collected at present are explained in the abstract. I suggest that the author add the purpose and significance of this technique to the summary section in a short language. Secondly,

[Figure]

I recommend that the Database records for individual projects could be presented in Table, and it also shows the collection status of water samples on the global map (like Fig.2), and clearly labels specific sites and projects. Thirdly, Can the explanation of the method be more detailed and specific? Can you make a clear illustration of the three ways of introducing new data to the database? I suggest that the author make a simple sketch so that everyone can understand it. Finally, Figure 3 is not clear. I recommend that the author redraw Figure 3. It's better to distinguish the color clear.

---

## Author Comment (AC1) · 21 May 2019

The authors would like to thank this reviewer for their kind remarks and their constructive comments. We will respond to a subset of comments here. First, we agree, the inclusion of a table instead of list for all of the projects will be useful for the reader. We plan to replace the list that is currently in the supplemental information with a large table containing the same information. However, due to the total number of projects and sampling sites, displaying this information clearly on one map figure may be challenging (Sites will plot over one another in most locations-check out the online database portal to get a sense of the site density). However, we are open to other ideas for

visualization of site and project distributions. We do agree that a flow chart indicating the methods for introducing data into the database may be helpful for readers to understand how the database is populated. This may be added as a second panel in Figure 1. Finally, to the last comment about Figure 3, does the reviewer suggest simply changing the colors to make this figure more clear? Or does the reviewer suggest a different type of figure? Again, thank you for your helpful review.

---

## Referee Comment (RC2) · Anonymous Referee #2 · 22 May 2019

The hydrogen and oxygen stable isotopes have been widely used to identify sources, transport pathways and phase-change processes within the water cycle in different fields. The technical note builds an open-access, global database of stable isotopic ratios of different water types. The work is very useful for data collecting and sharing. I think the note has provided detailed description about how to build, use and update the database. However, it would be helpful for readers to use the database if authors could consider the following two points. Firstly, describe the sample types more detail. For example, considering the phase of precipitation (rain, snow or mixed), the temperature of spring (hot or cold), the types or depths of groundwater (confined or unconfined, and shallow or deep). Secondly, give the homepage of the wiDB web interface in the

technical note.

---

## Author Comment (AC3) · 23 May 2019

The authors would like to thank this reviewer for their kind and constructive comments. Yes, we agree that more description of the different types would be useful for the reader, and will will plan to include a link to the homepage of the wiDB web interface in the document.

---

## Author Comment (AC4) · 28 Jun 2019

The authors thank the reviewers for their helpful comments. A summary of comments (italicized) is presented below along with the author's response. The responses include our intended changes to address these comments.
* * *
Reviewer 1:

1) *Only the importance of isotope research and the data collected at present are explained in the abstract. I suggest that the author add the purpose and significance of*

*this technique to the summary section in a short language.*

We will add a sentence or two describing the purpose and significance of this project to the summary.

2) *I recommend that the Database records for individual projects could be presented in Table, and it also shows the collection status of water samples on the global map (like Fig.2), and clearly labels specific sites and projects.*

We will plan to change the project list to an excel spreadsheet and include it as supplementary information. While we like the suggestion of a map that has each of the projects marked clearly, there is too much data to make this map useful. Instead, we direct those interetested to the database, where they can sort by project to see the spatial and temporal coverage of the projects interactively.

3) *Can the explanation of the method be more detailed and specific? Can you make a clear illustration of the three ways of introducing new data to the database? I suggest that the author make a simple sketch so that everyone can understand it.*

We will create a simple flow chart to help illuminate these processes, which we will include in an updated version of the manuscript.

4) *Figure 3 is not clear. I recommend that the author redraw Figure 3. It's better to distinguish the color clear.*

It seems that this reviewer primarily has an issue with the color scale used for this figure, as opposed to the layout. We will reprint this figure with an updated colorscale to improve the reader's ability to distinguish between the different sample types.

––––––––––––––––––––––––––––––––––––––––––––––––––––

*Reviewer 2:*

1) *Describe the sample types more detail. For example, considering the phase of precipitation (rain, snow or mixed), the temperature of spring (hot or cold), the types or*

*depths of groundwater (confined or unconfined, and shallow or deep).*

We will add a sentence or two describing how this information is stored within the database in the methods section, as these are attributes that are stored within the samples table.

2) *Give the homepage of the wiDB web interface in the manuscript.*

We will add the homepage of the wiDB to the manuscript.

———————————————————

---

## Author Response (AR1)

**"Technical note: A global database of the stable isotopic ratios of meteoric and terrestrial waters"**

**by Annie L. Putman and Gabriel J. Bowen**

*The authors thank Dr. Tian and two anonymous reviewers for their helpful comments. A summary of comments is presented below along with the author's response in blue italics. All page and line numbers refer to the marked up version of the text.*

**Handling editor, Dr. Fuqiang Tian:**

This manuscript introduced a database of isotopes in water. This database is very important because it is better than other existed databases on the aspect of spatial coverage, number of samples and water type. However, because of the importance of this database, we need to think carefully over the quality of data based on enough information. By now, I think the information provided in the manuscript is not enough and some issues should be clarified in more details.

*The authors thank Dr. Tian their helpful comments. We have worked to address the concerns of the editor, and hope our adjustments clarify the important points of the manuscript. Page and line numbers refer to the marked up version of the text.*

1) The description of method is not enough.
   a. How did you check the data contributed by scientists?
      *Within the text we state that we assume that data from peer-reviewed publications are quality checked as part of the publication and peer review process (P 5, L 16-17). However, we do check these data for 'reasonable isotope analysis values, dates and times, and geographic locations. Periodically, we manually check the whole database to ensure latitudes between -90° and 90° and longitudes between -180°and 180°. We check for reasonable water isotope analysis values by reference to the Global Meteoric Water Line, which describes the expected linear correlation between $d^{18}O$ and $d^2H$, combined with the expected ranges of each isotope in natural samples of various types.' (P 5, L 17-21). Finally, we caution database users to refer to the original publication/dataset in the case that any data 'appear inconsistent with expected values.' (P 5, L 22-23).*
   b. A large number of datasets are available from papers, but they may suffer from a wide range of metadata completeness as you say. How did you solve this problem and integrate the data with different quality together? Or did you remove the data without complete information?
      *We chose to include metadata in forms that are flexible and generalizable (Figure 1, panel a). For example, our collection date field includes both date and time, as well as time zone, so can handle sub-daily to multiyear collections. We have added text to clarify these issues, particularly with respect to the most critical metadata. (P 5, L 33 – P 6, L 4)*

c. Digitizing data and reconstructing metadata from figures are required in some cases. What kinds of figure have you reconstructed data from? And how about the software you used and its accuracy? Please clarify.

*Digitizing data for geographic information may mean approximating sample points from a figure in the manuscript overlaid on a google earth map. In other cases, the table including the data may come from a scanned pdf (in the case of older manuscripts) so the digitizing simply means transcribing the contents of the pdf table into an excel spreadsheet. This is typically done manually, although may be done in some cases aided by the pdf to excel spreadsheet conversion tool available from Adobe Acrobat Pro. We have added text for clarification. (P 4, L 10 – 14).*

2) The database is not displayed sufficiently in the manuscript.

a. Only the codes for processing are available (some of the link are not accessible, e.g., the xlsx format files in the 'reports_example_files' folder), but the database itself is not accessible for reader. Considering the database is not published yet, at least some small parts of the database could be provided for example, so that we can easily understand what the database looks like and how to use it.

*This is a misunderstanding, due in part because we did not provide the link to the database in the text. Apologies for the oversight. The database is indeed published, and is accessible at:*

*http://wateriso.utah.edu/waterisotopes/pages/spatial_db/SPATIAL_DB.html*

*At the request of the reviewers, we have also added this link in the introduction of the wiDB, in the Introduction section as well as in the section titled "Usage notes and further comments". (P 3, L 4 and P 6, L 8)*

b. Some important properties of the database are not displayed.

   i. The authors mentioned some of the data are not available, and only public data are permitted downloads. How many are they?

   *These details, included as number and percentage of samples is now included in the Database records section. (P 4, L 28-30)*

   ii. Please clarify how many data are from scientists' contribution and peer-reviewed paper.

   *74.2% of samples in the database come from agency or network datasets. Interested readers can use the table in the supplemental information to sort and sum the data in different ways. (P 4, L 30 -32).*

   iii. Please display the spatial distribution of different sample types, because it can be quite different according to some existed database.

   *While we like the idea of being able to explicitly show the different sample types on a map, the map would likely contain too many points to be interpretable. Instead, we suggest that readers who are interested in the distributions of different samples types visit the web portal and use the filtering tools provided there. To clarify this point, we have added this additional information to the section "Database records" (P 5, L 1-2).*

iv. Again, did you remove the data without complete metadata? If not, the data will have different levels of quality. Please show the number of data with different quality. Also, please show the number of the data you constructed from figures.

*As explained in the response above, we do tolerate data with some missing information. However, the most important information (geographic, water type, collection time) is nearly always present (99% of the time) if we preserve data. This detail is included in the Technical validation section. (P 6, L 3-4).*

**Reviewer 1:**

1) Only the importance of isotope research and the data collected at present are explained in the abstract. I suggest that the author add the purpose and significance of this technique to the summary section in a short language.

*We have added "The motivation to develop this database comes from water isotope ratios utility in identifying sources, transport pathways, and phase-change processes within the water cycle, which can be used for hydrologic, forensic, ecologic, and hydroclimatic investigations." to the conclusion. (P 7, L 4-6)*

2) I recommend that the Database records for individual projects could be presented in Table, and it also shows the collection status of water samples on the global map (like Fig.2), and clearly labels specific sites and projects

*We have replaced the list of projects in the supplemental with an excel spreadsheet which contains the same information but is more easily searchable and sortable, which is included as a supplement. However, for the best opportunities to search the available data, we recommend users visit the database portal.*

*While we like the idea of a static map of all sites and projects, the amount of data that we'd need to display would render the map difficult for a reader to interpret. Thus, we suggest that readers interested in more detail about any project visit the wiDB portal where the data may be sorted by project, type, and time range, among other parameters. We have added this additional information to the section "Database records" (P 5, L 1-2).*

3) Can the explanation of the method be more detailed and specific? Can you make a clear illustration of the three ways of introducing new data to the database? I suggest that the author make a simple sketch so that everyone can understand it.

*We have created a flow chart, which has been added as panel (b) to Figure 1. This panel is referenced in the text (P 3, L 30), and will assist readers in understanding the sources of data, and how they are passed to the wiDB.*

4) Figure 3 is not clear. I recommend that the author redraw Figure 3. It's better to distinguish the color clear.

*We have changed the color scheme and distribution of the different sample types to aid in the interpretation of Figure 3 by the reader.*

**Reviewer 2:**

1) Describe the sample types more detail. For example, considering the phase of precipitation (rain, snow or mixed), the temperature of spring (hot or cold), the types or depths of groundwater (confined or unconfined, and shallow or deep).
   *We have added the sentence "The samples table also includes pertinent information for specific water types, like the phase of precipitation (solid, liquid), and depth of groundwater sampling." to the Methods section. (P3, L 23-25)*

2) Give the homepage of the wiDB web interface in the manuscript.
   *The URL of the wiDB is now included with the introduction of the wiDB, in the Introduction section as well as in the section titled "Usage notes and further comments". (P 3, L 4 and P 6, L 8)*

**List of relevant changes:**

Abstract (P 1, L5 -6) and Database records (P 4, L 22 – 30): All statistics of data included in the database (number of samples, sites, projects and water isotope analyses) were updated from the April 2019 numbers to the July 2019 numbers.

P 3, L 4 and P 6, L 8: Included the wiDB URL

P 3, L 23 – 25: Further explanation of specific metadata included in *Samples* table.

P 3, L 30: Reference to new panel of Figure 1, which shows the three ways data was introduced into the database.

P 4, L 10 -14: More detail was included about how data was digitized and how sample locations were reconstructed if no geographic data was included.

P 4, L 28 – P 5, L 2: More descriptive information about the database was added.

P 6, L 1 – 4: Further information about handling of missing data, and specification of critical metadata.

P 7, L 4 – 6: Reiteration of motivation for development of database.

[revised manuscript text omitted]

---

## Editor Decision (ED1)

This manuscript introduced a database of isotopes in water. This database is very important because it is better than other existed databases on the aspect of spatial coverage, number of samples and water type. However, because of the importance of this database, we need to think carefully over the quality of data based on enough information. By now, I think the information provided in the manuscript is not enough and some issues should be clarified in more details.

The description of method is not enough.
- How did you check the data contributed by scientists?
- A large number of datasets are available from papers, but they may suffer from a wide range of metadata completeness as you say. How did you solve this problem and integrate the data with different quality together? Or did you remove the data without complete information?
- Digitizing data and reconstructing metadata from figures are required in some cases. What kinds of figure have you reconstructed data from? And how about the software you used and its accuracy? Please clarify.

The database is not displayed sufficiently in the manuscript.
- Only the codes for processing are available (some of the link are not accessible, e.g., the xlsx format files in the 'reports_example_files' folder), but the database itself is not accessible for reader. Considering the database is not published yet, at least some small parts of the database could be provided for example, so that we can easily understand what the database looks like and how to use it.
- Some important properties of the database are not displayed.
  - The authors mentioned some of the data are not available, and only public data are permitted downloads. How many are they?
  - Please clarify how many data are from scientists' contribution and peer-reviewed paper.
  - Please display the spatial distribution of different sample types, because it can be quite different according to some existed database.
  - Again, did you remove the data without complete metadata? If not, the data will have different levels of quality. Please show the number of data with different quality. Also, please show the number of the data you constructed from figures.

---

## Author Response (AR2)

**Response to editor and reviewers for**

**"Technical note: A global database of the stable isotopic ratios of meteoric and terrestrial waters"**

**by Annie L. Putman and Gabriel J. Bowen**

*The authors thank Dr. Tian and two anonymous reviewers for their helpful comments. A summary of comments is presented below along with the author's response in blue italics. All page and line numbers refer to the marked up version of the text.*

**Handling editor, Dr. Fuqiang Tian:**

I agree with the Referees' comments. The manuscript is almost ready to publish. Please consider the minor comments by the Referee #2 and submit the revised manuscript. Thanks.

*Thank you for the thorough review. Please find our response to Reviewer #2's comment below.*

**Reviewer 2:**

The manuscript has been improved. Regarding the first issue I raised in the first review, I suggest adding the temperature of groundwater, spring and surface water (river, lake etc.) in the database for extending the application scope of the database. Water temperature is an important environment tracer. For example, it can be used to clarify the interaction between groundwater and surface water. It is also helpful to analyze the process of groundwater circulation.

*We understand this reviewer's interest in having these specific ancillary data contained in the database. As the database is currently configured, we do not have this information stored. In my experiences organizing and processing water isotope data, water temperature information is not often recorded or made available by the study authors (exceptions include the USGS groundwater data and other similar agency data). Furthermore, we do not store other potentially useful quantities like pH, dissolved carbon carbon-14 age, general ion chemistry, etc.*

*We realize that not storing these ancillary data additional information may limit the utility of this data for specific projects. However, users are able (and encouraged) to make use of the provenance information that we provide to gather this ancillary data themselves, if their specific question requires this information and it is available.*

*As the Earth Science community continues to make progress in the ways we store and share data, there may be opportunities for further expansion of the types of data stored by the wiDB, or linking of this database with others that hold those types of information. However, at this time, adding this field (and populating it) in the database is beyond the scope of this work and we have made no changes to the manuscript.*

**List of relevant changes:**

No changes were made to the manuscript.